# Microscale Templating of Materials across Electrospray Deposition Regimes

**Michael J. Grzenda** [1], **Maria Atzampou** [2], **Alfusainey Samateh** [3], **Andrei Jitianu** [3,4], **Jeffrey D. Zahn** [2] **and Jonathan P. Singer** [1,5,*]

1. Department of Materials Science and Engineering, Rutgers University, Piscataway, NJ 08854, USA
2. Department of Biomedical Engineering, Rutgers University, Piscataway, NJ 08854, USA
3. Department of Chemistry, Lehman College, of the City University of New York, New York, NY 10468, USA
4. Ph.D. Program in Chemistry and Biochemistry, The Graduate Center of the City University of New York, 365 Fifth Avenue, New York, NY 10016, USA
5. Department of Mechanical and Aerospace Engineering, Rutgers University, Piscataway, NJ 08854, USA
* Correspondence: jonathan.singer@rutgers.edu

**Abstract:** Electrospray deposition (ESD) uses strong electric fields to produce generations of monodisperse droplets from solutions and dispersions that are driven toward grounded substrates. When soft materials are delivered, the behavior of the growing film depends on the film's ability to dissipate charge, which is strongly tied to its mobility for dielectric materials. Accordingly, there exist three regimes of electrospray: electrowetting, charged melt, and self-limiting. In the self-limiting regime, it has been recently shown that the targeted nature of these sprays allows for corona-free 3D coating. While ESD patterning on the micron-scale has been studied for decades, most typically through the use of insulating masks, there has been no comparative study of this phenomenon across spray regimes. Here, we used test-patterns composed of gratings that range in both feature size (30–240 μm) and spacing ($^{1}/_{3}$x–9x) to compare materials across regimes. The sprayed patterns were scanned using a profilometer, and the density, average height, and specificity were extracted. From these results, it was demonstrated that material deposited in the self-limiting regime showed the highest uniformity and specificity on small features as compared to electrowetting and charged melt sprays. Self-limiting electrospray deposition is, therefore, the best suited for modification of prefabricated electrode patterns.

**Keywords:** electrospray deposition; electronic packaging; fabrication; microtechnology; thin films

## 1. Introduction

As microscale manufacturing grows in scale, the cost of production can be greatly reduced through manufacturing techniques that operate at ambient pressures and low temperatures. While electrosprays have long been a keystone of certain large-scale manufacturing industries such as paint coating and agriculture, electrospray deposition (ESD) has yet to reach broad industrial-scale application. This distinct regime of electrospray operates at low flowrates and relies on only electrostatic forces to deliver material. ESD has several unique capabilities for thin film [1–4] and nanoparticle [5–8] processing that are more relevant now than ever. However, further understanding and demonstration of the capabilities of ESD are necessary before employing it in, for example, the creation of dielectric coatings, hermetic barriers, and the deposition of active sensor materials. More specifically, this manuscript explores the potential for ESD to be combined with microfabricated templates without the need of a stencil mask through understanding and leveraging of charge dissipation mechanisms.

During ESD, a strong electric field is applied to a capillary containing a solution or dispersion. This causes the exiting fluid to undergo electrostatic breakdown and produce generations of charged monodisperse droplets that deliver their payloads to a grounded

target [9]. For ESD of non-conductive soft matter, we have categorized this behavior into three regimes that depend on the ability of the deposited film to dissipate charge [10]. If the deposited film maintains a certain level of mobility and low viscosity, it will thin to increase surface area and conduct charge to uncoated regions, which is called electrowetting. While spraying liquids or solid materials above their glass transition temperature ($T_g$) is an obvious way to maintain mobility, it has also been observed that higher temperatures can induce greater mobility by increasing the solvent absorption, which, in turn, decreases the $T_g$. The charged melt regime also depends on mobile material, but in this case, the film's higher viscosity favors instabilities to dissipate charge. Due to the dependence on mobility, these electrospray regimes exist in a balance, and it has been observed that electrowetting materials can transition to charged melts as the material thickness increases. Further, if the surface affinity to the target is poor, either regime can result in dewetted films. On the completely opposite end of the spectrum is the self-limiting (SL) regime where glassy materials below their effective $T_g$ are sprayed in a phenomenon we have named self-limiting electrospray deposition (SLED). In the SL regime, charge is unable to dissipate from the growing film, and this charge build-up eventually results in the repulsion of nearly all the incoming material. This regime also transitions to the charged melt regime as $T_g$ is approached through either increased temperature or blending with solvent vapor or another material, as the charge is better able to dissipate through mass transport [11]. One of the main advantages of SLED is the ability to conformally coat conductive/charge mobile surfaces, and much of our prior work has focused on 3D, corona-free coatings [12]. However, the ability to target micron-scale 2D patterns is also highly valuable, but in need of further characterization.

While several studies have patterned electrosprayed materials simply by controlling local electric fields [13,14], the typical approach has been through some form of masking, with authors depositing nanomaterials (both organic and inorganic), biomaterials, and polymers [15–22]. In one of the earliest examples of patterning with ESD, Buchko et al. used a shadow mask for depositing arrays of polypeptides, with the term shadow mask implying that the mask was conductive, and the substrate and mask were equally coated like a stencil [23]. It was Morozov and Morozova who first described the use of an insulating mask for the deposition of arrays of DNA material with the observation that the electrospray quickly traps charge on the mask, leading to a repulsion that focuses the spray onto conductive regions [24]. Importantly, this focusing increases the deposition thickness relative to an unmasked substrate. Hu et al. also observed that focusing could even be achieved without a mask when spraying block copolymers onto glass using a grounded needle and a high voltage grating beneath the substrate to direct the spray [14]. Interestingly, their (likely charged melt) spray was observed to spread beyond the width of the buried grating.

While material focusing can be useful, especially when attempting to achieve small linewidths, we must also consider spray uniformity. For unpatterned, non-SLED sprays, deposition tends to lead to a peaked distribution. On the other hand, focusing depends greatly on geometry, meaning that the spray of complex 2D patterns will lead to large variations in the amount of material deposited, which can be especially problematic when manufacturing multi-layer materials. This was encountered by Morozov and Morozova, who tried to counteract the effects either by moving the spray platform or by adding a shield to the spray environment [24]. Though successful to a degree, their array features were all of equal size, so it is unclear if this effect would carry over to more complex geometries.

Further, work from our group has shown that even materials sprayed in the SLED regime can be impacted by feature size. In a recent study, focused laser-spike (FlaSk) dewetting was used to pattern circular holes of varying sizes on photoresist-coated silicon wafers as templates [22]. After spraying with polyvinylpyrrolidone in the SL regime, it was observed that larger features tended to have thicker coatings, and that all features had coatings that were thicker than an unmasked substrate. This is curious in light of our initial work that showed that thickness in the SL regime was dependent only on

electric field strength, and that below the SL thickness, total sprayed mass was the only determining factor. However, at the time, it was observed that deposited materials were slightly collapsed, indicating that excessive solvent vapor may have been generated due to focusing from the template. As mentioned earlier, increased solvent absorption can lead to charge dissipation and electrowetting/charged melt behavior. These effects can, therefore, be mitigated in three ways: (1) the total patterned area can be increased, (2) the flowrate can be reduced, and (3) the strength of the dielectric layer can be reduced to decrease the effect of focusing. Regarding the third point, Zhu and Chiarot observed that thinner photoresist layers allowed for deposited nanoparticles to strike closer to recessed feature edges, indicating a reduction in the focusing effect [15].

Here, we aim to elucidate the performance of all three spray regimes by spraying substrates composed of silicon chips patterned with Ti/Pt on top of an insulating layer of Parylene. Furthermore, we aim to quantify the interaction of complex 2D geometry with ESD through the use of a metal pattern that includes multiple gratings of varying sizes and spacing. A MATLAB script is used to extract values for average deposition height, volumetric density, along with a simple figure of merit that characterizes the specificity of the spray to the template.

## 2. Materials and Methods

### 2.1. Materials

Polystyrene (35 kDa) and 2-Butanone (>99%) were purchased from Sigma Aldrich (St. Louis, MO, USA) and were used as received. Polystyrene (PS) was added to butanone at a mass loading of 1 wt.% and left on a roller overnight. A "melting gel" material was used to demonstrate a fully liquid, electrowetting spray. Melting gels (MGs) are oligomeric silsesquioxanes (synthesized via a sol–gel process described elsewhere [25]) that show thermoplastic-like behavior above their $T_g$, but cross-link into hybrid organic/inorganic glasses when heated past their consolidation temperature [26]. The melting gel used here was composed of 65% methyltriethoxysilane (MTES) and 35% dimethyldiethoxysilane (DMDES) (in mol%) and had a $T_g$ below 0 °C. The MG was diluted using absolute ethanol to 1 wt.% MG for spray. All sprays were performed on Ti/Pt patterns on 4–5 μm thick Parylene insulation. These substrates were made in-house via photolithography and acetone lift-off using 4-inch silicon wafer substrates.

### 2.2. Feature Test Patterns

The feature test patterns were designed to assess a series of gratings that range in both grating width and spacing (Figure 1). The gratings were attached to a $1 \times 1$ cm$^2$ grid used as a grounding path connected to a 0.5 cm$^2$ grounding pad. Every individual grating filled an approximate width of 1.5 mm and the features had a length of 1 mm. On the interior of the grounding grid, the feature width, referred to as the feature size, increased in multiples of 15 (15/30/60/120/240 μm). Around the exterior, the feature sizes increased in multiples of 20 (20/40/80/160 μm). Along each row, the feature size was constant, but the gap between each finger increased as a function of the feature size ($^1/_3$x, 1x, 2x, 3x, and 9x). For example, the 30 μm row had gaps of 10, 30, 60, 90, and 270 μm. For this work, we focused on the 30/60/120/240 μm features. The 20 μm features were included to contrast the 30 μm features due to their similar size but different spatial location on the test pattern. However, for both these smaller feature sizes, the $^1/_3$x gratings were excluded because they were too small for the profilometer to measure.

### 2.3. Electrospray Setup

Strong electric fields were applied via alligator clips to a vertically suspended 30-gauge needle and a stainless-steel ring (ID 2 cm) using high-voltage power supplies (Acopian, P012HA5M, Easton, PA, USA) to achieve electrospray. The ring was placed 1 cm above the needle tip and acted to stabilize the electric field and focus the spray. The voltage on the needle was held at a constant 6 kV DC, while the voltage on the ring varied from 3 to 5.5 kV

DC to maintain stable cone jet sprays. The solution was pumped to the high-voltage vertical needle using a syringe and Teflon tubing using a syringe pump at a rate of 0.1 mL/hr. During sample fabrication, the spray was first stabilized on a grounded, blank, 4-inch Si wafer (p-type; 0–100 $\Omega$ cm), stacked on top of a steel plate and hot plate, and located 6 cm below the needle tip. Once the spray was stabilized, the test pattern was placed in the center of the spray and allowed to coat for 30 min. Separately, the grounding pad of the test pattern was taped with carbon tape to a 2-inch wafer (p-type; 0–100 $\Omega$ cm) to create a grounding path. This 2-inch wafer was able to interface directly with the grounded 4-inch wafer. While the PS sprays were sprayed in ambient atmosphere (30% humidity/22 °C), the MG was sprayed in a low-humidity chamber (21% humidity/25 °C) due to its moisture sensitivity. For the PS, two sprays were performed, one at a hotplate temperature of 30 °C, which is in the SL regime, and the other at 100 °C, which is near or above $T_g$ due to solvent swelling. The MG was sprayed at 35 °C to completely ensure it was sprayed as a liquid. After spraying, the MG was consolidated overnight at 150 °C to fully crosslink. Although SLED sprays are highly porous, the PS was densified via solvent swelling using a brief exposure to butanone vapor to allow for profilometry scanning.

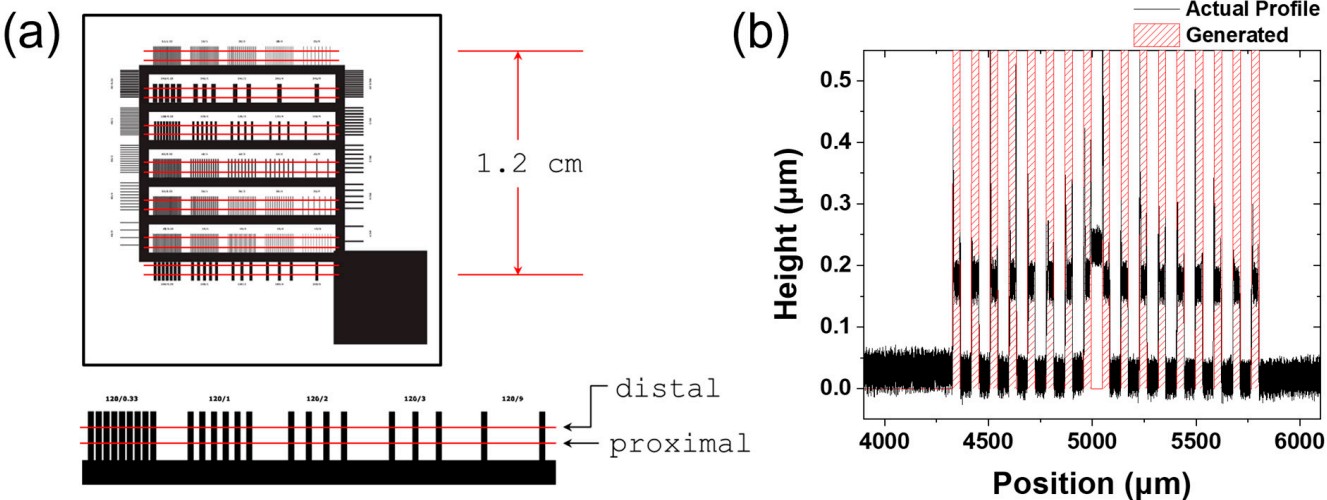

**Figure 1.** (**a**) (top) Full feature test pattern and (bottom) single row demonstrating the distal/proximal scans. (**b**) Actual uncoated test pattern profile (black) compared to calculated binary profile (shaded red). An error can be observed in the center of the grating that has been corrected in the model.

### 2.4. Measurement and Analysis

The test patterns were measured using a KLA Tencor P-7 Stylus Profilometer (Milpitas, CA, USA). Each feature was scanned twice, with a 0.5 mm spacing between scans (Figure 1). We differentiate the scans by referring to them as distal or proximal with respect to the grounding grid. Scans were performed with a 100 μm/s scan speed, 200 Hz sampling rate, and a 0.5 μm step size. The stylus had a 2 μm radius and contacted the surface with a pressure of 2 mg per stylus area.

Scans were analyzed using a MATLAB (R2022a, Mathworks®, Natick, MA, USA) script. A blank test pattern was scanned to create a baseline. To remove noise and standardize the analysis, the average feature size (width) was measured from these scans and used to create a binary model of the gratings (Figure 1). The model gaps were calculated for each grating by subtracting the average measured feature size from the total spacing of a gap plus feature size. The measured average width was a few microns larger than the actual feature size due to the stylus diameter. During analysis, the script aligned the collected profile scan with the binary data file (Figures S1–S3, see Supplementary Materials) and calculated 3 parameters: (1) height ($H_g$) is the average height of the profile directly on top of the features for a given grating; (2) density ($\rho_g$) is the total amount of material deposited, in area, divided by the length of the features available; (3) specificity ($\sigma_g$) is the amount of

material on the features divided by the total amount of material deposited, both as areas. The calculation is accomplished through rectangular approximation and array operations:

$$H_g = \frac{\sum(h_p \circ m)\Delta x}{\sum(m)\,\Delta x} \tag{1}$$

$$\rho_g = \frac{\sum(h_p)\Delta x}{\sum(m)\,\Delta x} \tag{2}$$

$$\sigma_g = \frac{\sum(h_p \circ m)\Delta x}{\sum(h_p)\Delta x} \tag{3}$$

The height, density, and specificity are given a subscript ($g$) to denote that each parameter is calculated for every grating. Meanwhile, $h_p$ is the array containing the profile height for the grating, $m$ is the array containing the binary model, $\Delta x$ is the step size, and ($h_p \circ m$) indicates element-wise multiplication. Although $\Delta x$ cancels, we leave it here for clarity regarding the rectangular approximation. Despite having the same units, the height is best understood in terms of length (μm) and the density as an area per length (μm$^2$/μm). On the other hand, the specificity is unitless and ranges from 0 to 1, serving as a representation of how much of the sprayed material landed on the template. A visual representation of these parameters is given in Figure S4. Thanks to the similarity between these equations, the resulting relationship exists that is utilized later:

$$\sigma_g = \frac{H_g}{\rho_g} \tag{4}$$

During analysis, a cutoff value was added so that noise from the profile baseline was not included. These cutoff values were within range of the uncoated feature height (~0.15 μm), so the area of the actual feature was not considered in the density calculation. The analysis was also run with a simulated coating file where the entire profile consisted of $h_p = 1$. This represents a completely indiscriminate coating and is useful for identifying biases in our analyses.

### 3. Results and Discussion

To determine the importance of the ESD regime in template interactions, we employ several materials, real and simulated, that are expected to deliver highly different behaviors. As mentioned, the simulated data are used to model an indiscriminate coating with a uniform height (labeled $H = 1$ on graphs as we also scale it after analysis), such as would be deposited by, for example, spin coating, highly far-field spray, or vapor deposition. This "material" is useful for identifying biases in our metrics where smaller gaps result in lower densities and higher specificities. Real materials are compared to these biases to ensure observed trends are due to electrostatic phenomena. The melting gel (MG), showing liquid electrowetting behavior, demonstrates the least amount of charge build-up and, therefore, the least amount of repulsion. Without repulsion, a central area receives the most spray. This creates a gradient in spray thickness that, as is shown, almost entirely dominates the behavior observed. The 100 °C polystyrene (PS), demonstrating charged melt behavior, accumulates some charge, largely avoiding the formation of a gradient. However, its ability to spread and form instabilities dissipates some of the charge, leading to higher densities, lower specificities, as well as greater heights on large features. It is also seen that distal areas of the feature are more susceptible to this effect. Finally, 30 °C PS demonstrates a behavior entirely consistent with SLED, with charge accumulation redirecting the spray to uncoated regions. There is no thickness gradient or trend toward metric bias. Its density and height are consistent with the expectation of uniform growth on all features, regardless of size. Meanwhile, its specificity on small features consistently outperforms other materials.

### 3.1. Density and Height

Before investigating geometry, it is important to demonstrate where macroscopic thickness gradients influence our results. Figure 2 shows a surface map of the test pattern, with the density of each grating plotted based on its relative position on the test pattern while Figure 3 shows the same data plotted by feature size and spacing. MG, unlike the PS sprays, correlates with the position on the test pattern as opposed to feature size and gap. It is apparent that the corner of the test pattern receives a greater amount of spray than the rest of the pattern. As mentioned, the high mobility of the liquid dissipates charge and prevents the repulsion needed to counteract a thickness gradient. From the simulated coating, we can see that there is some bias in the metrics (bottom right) showing a similar trend. It is much easier to have a higher density coating for larger gaps simply because there is less feature surface available. However, this is not the whole story for MG. If we look at the actual profilometry data, we see that the heights of the MG features increase as we move across the test pattern, even within the same grating, which can only be explained by a gradient (Figure S5a).

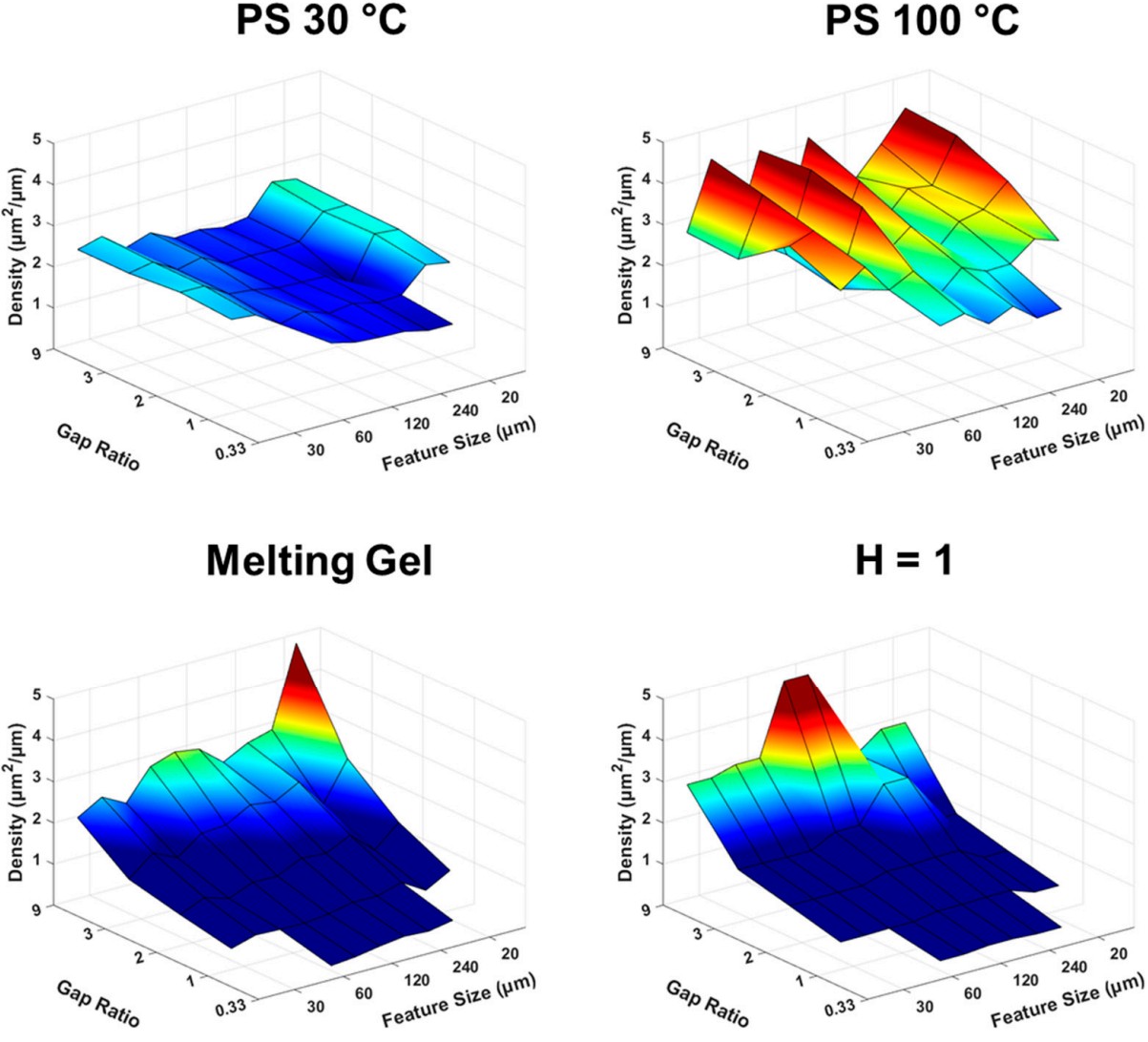

**Figure 2.** Surface maps of grating density plotted by their relative position on the chip. Note that 20 and 30 μm features are positioned on opposite ends. The simulated indiscriminate coating (*H* = 1) has been scaled to fit on the same axis as the real materials, but the actual values are arbitrary.

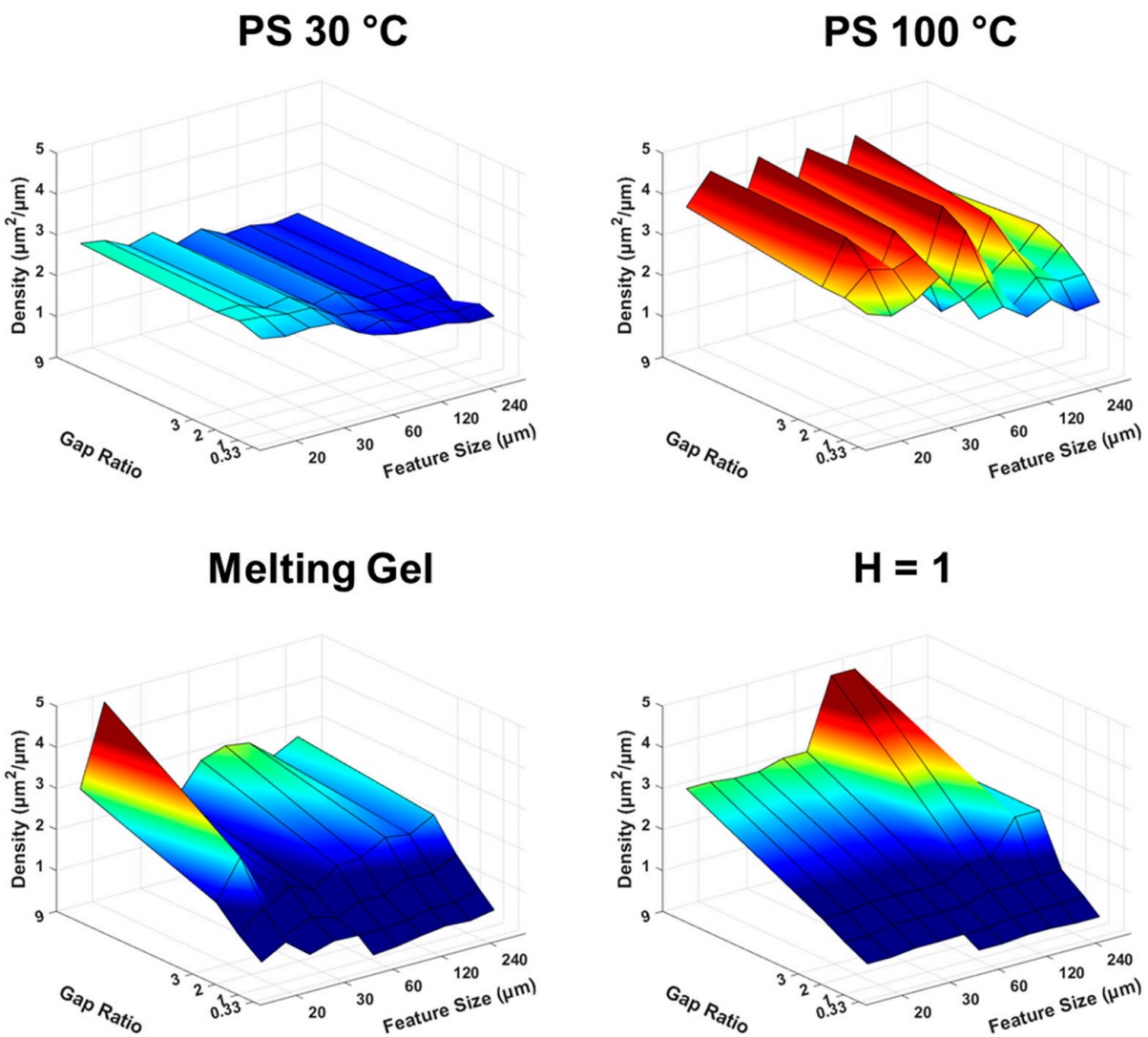

**Figure 3.** Surface maps of grating density plotted according to their gap ratio and feature size.

Due to this gradient, trends in the MG height data are more anecdotal than quantitative. Comparing the 20 μm large gap features to the 240 μm large gap features, we see an excellent example of the distinction between density and height (Figure S5b). There is clearly much more total material on the 20 μm features, even when there is much less total available surface as seen for the 9× gap. However, only the top left grating (20 micron/3× gap/distal) comes close to the heights seen for the 240 features. Unfortunately, without a single 20-micron feature to compare to, it is difficult to tell if this pattern is due to the feature size, the presence of multiple features, or various focusing or wetting effects.

Moving onto the PS sprays, we do not see evidence for a thickness gradient with relative grating position (Figure 2). While the 100 °C increases in density with increasing gap, this is not seen in the profile data as we saw with the MG, indicating that the trend is due to metric bias. On the other hand, for both sprays, we see a slight trend with respect to feature size with higher densities on smaller features (Figure 3). Importantly, this trend differs from the biases represented by the simulated data, indicating that it is a real effect. We also see that, on average, the 100 °C spray has a higher density than the 30 °C does. However, the 100 °C also has greater variation, appearing to trend with the scan's local position within the grating (distal/proximal) and not with the position on the test pattern.

From the height data, we see further variation between the 30 °C and 100 °C samples (Figure 4a). For 100 °C PS, we see that a definite trend exists for feature size. Demonstrating

the lack of thickness gradient, we see that the 30 μm features appear slightly taller than the 20 μm features do, despite the 20 μm feature's close physical proximity to the tallest features, 240 μm (Figure 4b). Looking at all the data points, we see that the 30 °C PS height barely changes with feature size. Averaging these data points, we obtain a height of 1.51 μm ± 0.10 for 30 °C PS features. This differs from our prior results that showed increasing deposition thickness on larger features [22]. This is likely the effect of a reduced spray delivery rate from a combination of (1) distance, (2) flowrate, and (3) overall pattern area, which, for this pattern, is ~0.9 cm$^2$ and, for the earlier work, was sub-mm$^2$. For this reason, the current result can be more fully considered SLED. This is reinforced by the templated SL thickness also differing from the SL thickness observed for PS on a bare wafer, which is ~2.5 μm [10,11]. This indicates that the presence of the charged mask makes the exposed template a less favorable target than the supporting ground, reducing the thickness needed to repel incident spray. This is similar to results reported by Kingsley et al., which indicated that small targets receive very little incident spray when depositing in the SL regime [21].

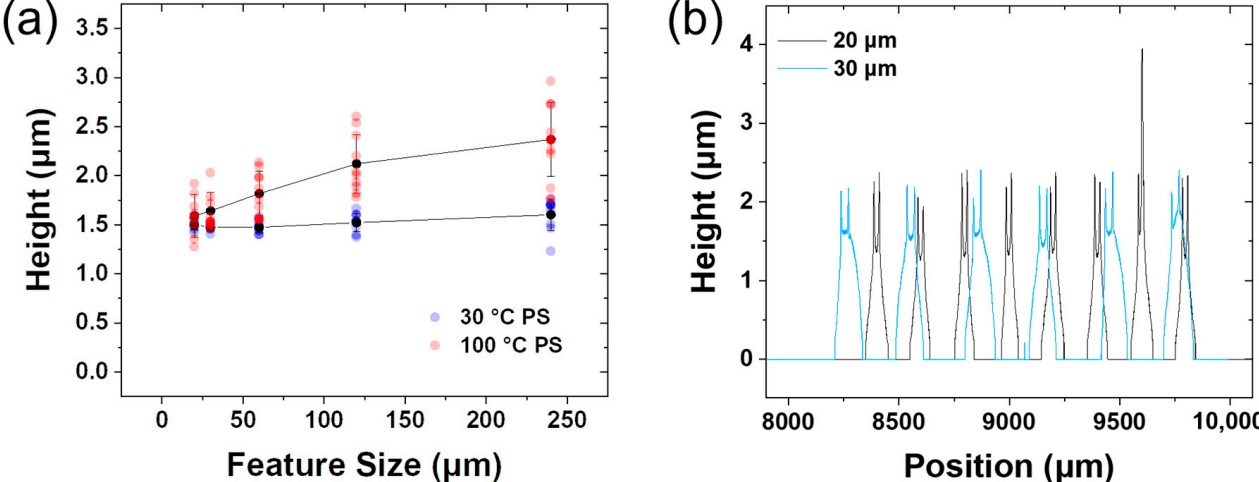

**Figure 4.** (**a**) Height data from the PS sprays; (**b**) Overlay of the 20 μm and 30 μm/9× gap distal features for the 100 °C spray showing a lack of a positional thickness gradient.

To verify and interpret these results, we overlay the profiles for the 30, 60, and 240 μm features for the 30 °C PS proximal as well as the 100 °C PS proximal and distal scans (Figure S6). To compare the profile shape, each feature is centered at zero, and the position data are normalized by the feature size. The profiles show a characteristic shape, with peaks near the feature edge. While profilometry can produce artifacts near sharp edges, as seen in Figure 1b, the raised edges seen here are too broad to fit this description. Instead, this is likely the effect of the surrounding insulating areas focusing material onto the feature edges. The hypothesis is supported through finite element method simulations, which show the electric field lines preferentially directing material toward the edge of the feature when there is a moderate amount of surface charge on the surrounding insulation (Figure S11). It is also important to remember that these coatings are densified, and the actual effect may be even more prominent for undensified material. Comparing between feature sizes, of particular note is the change in shape seen between the 60 and 240 μm features, which is most easily discernable in 30 °C PS. As the edges increase in thickness, they also increase in repulsion. It appears that this repulsion either prevents further deposition on the center of smaller features or acts to focus material onto the centers of larger ones. Comparing across materials, we can gain a much clearer understanding of the height results seen in Figure 4a. There is clearly much greater uniformity in the 30 °C PS profile compared to the 100 °C PS, which explains the relative uniformity of the 30 °C PS height data. While the central regions are increasing in height, this result is tempered due to the small size of the difference and the uniformity of the edge heights. For the 100 °C PS, we can see that

the proximal profiles are more uniform than the distal. For both these profiles, the 30 μm features appear to have segregated into two bands, one with a concave central region and another that is relatively flat. This is likely due to the proximity of surrounding features, with smaller gaps allowing for more blending and spreading, creating the flatter band. For the 60 and 240 μm features, there is more room to spread on an individual feature and the variability increases. For the 240 μm features in particular, the curvature becomes distinctly convex, and the central region can grow far beyond the height of the edge peaks.

### 3.2. Specificity

The specificity of all materials and the simulated data are plotted on surface maps in Figure 5 according to their feature size and gap ratio. For all real materials, the specificity values indicate that it is clearly more difficult to target smaller features, as would be expected. We can also see that our metrics do not bias the results in this direction; rather, the simulated data show us that smaller gap ratios produce a bias toward higher specificity values due to the greater available surface. To probe the specificity further, we turn to a comparison plot that includes both specificity and density, shown in Figure S7 for all materials. Figures 6 and 7 show individual plots of specificity versus the inverse of density with color maps denoting both gap ratio and feature size, respectively. Beginning with the uniform data, we see the justification for the transformation: the relationship between specificity and the inverse of density is now linear, which can be explained by Equation (4) where $H_g$ is equal to 1. For this case, specificity and density are mainly controlled by the amount of the feature available, which is generally controlled by the gap. With the amount of space per grating being relatively constant, larger gaps mean less available surface and vice versa. This explains why the gap-ratio color map is completely ordered.

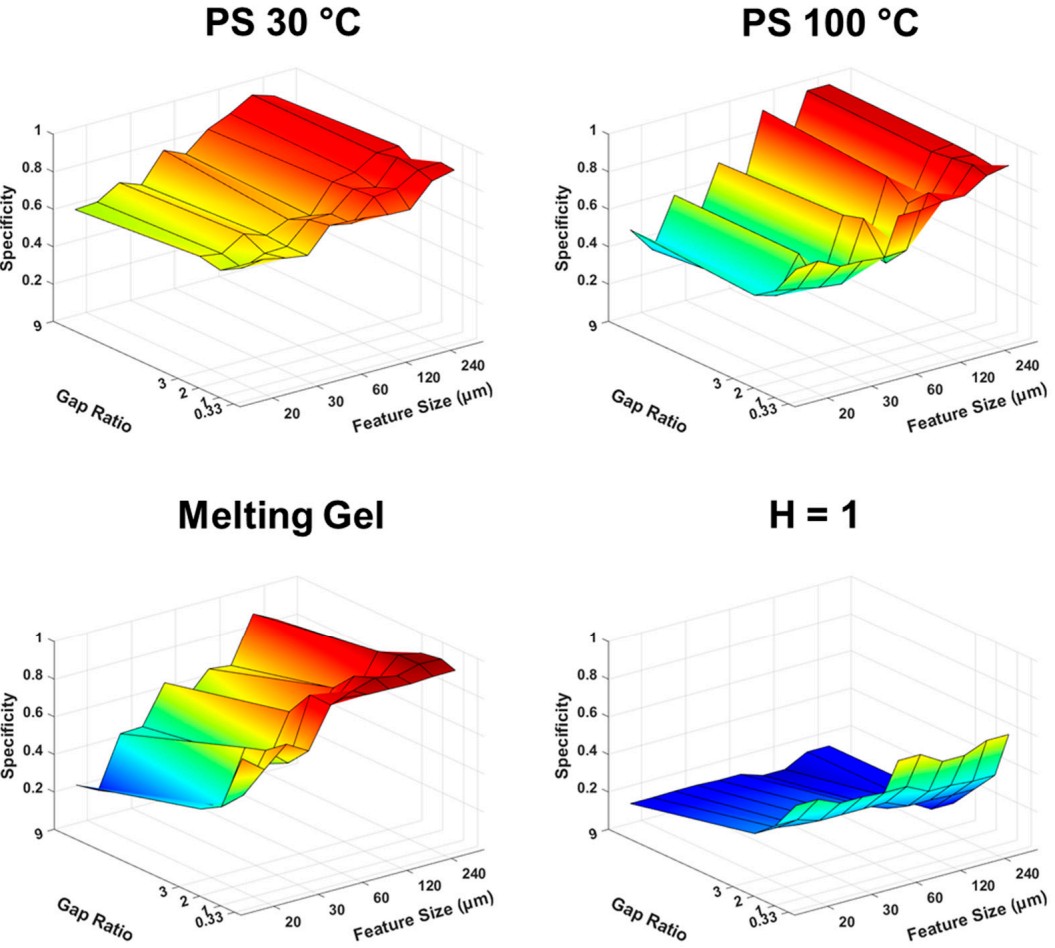

**Figure 5.** Surface maps of specificity graphed according to the feature size and gap ratio.

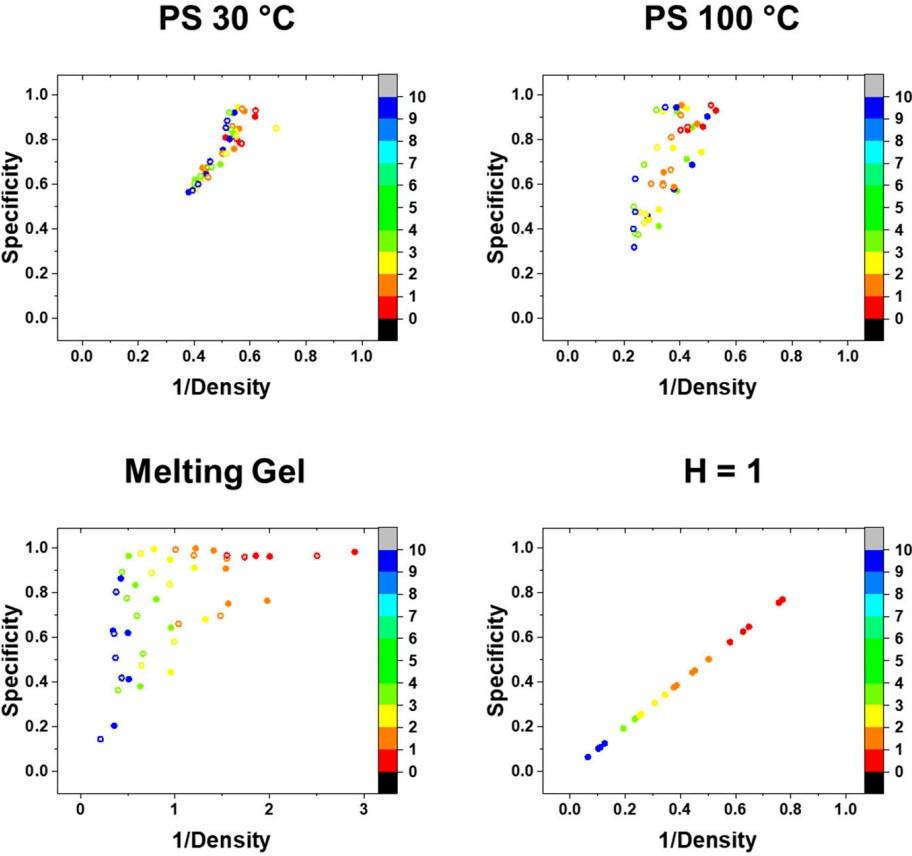

**Figure 6.** Specificity vs. 1/Density with color maps denoting gap ratio.

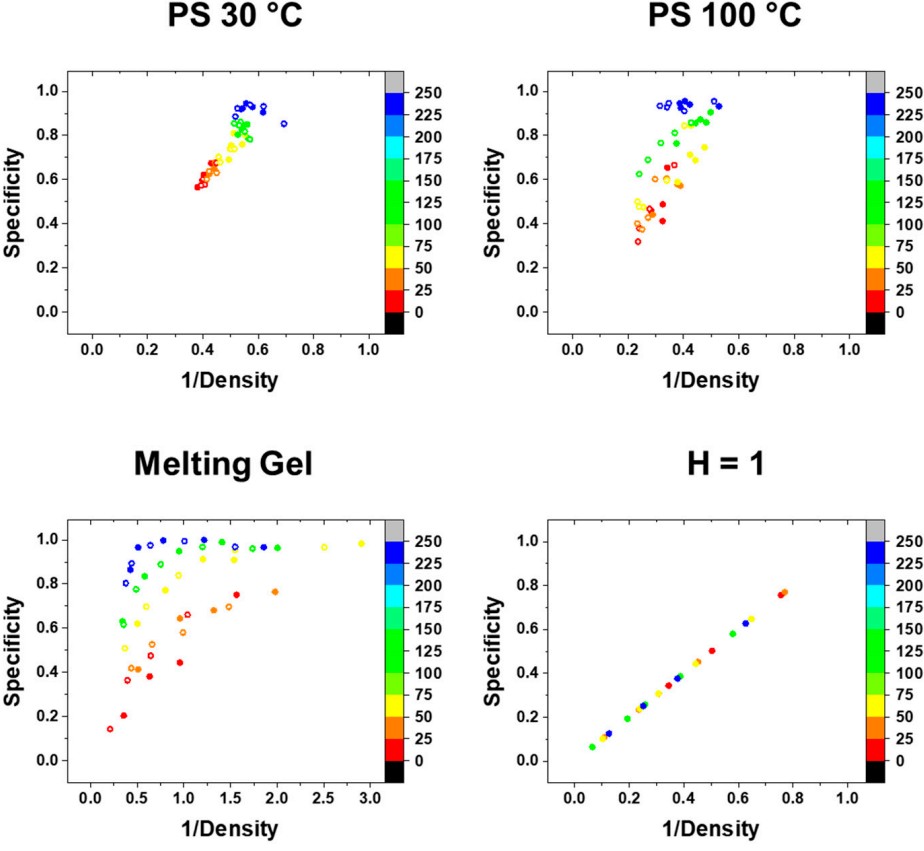

**Figure 7.** Specificity vs. 1/Density with color maps denoting feature size (μm).

Importantly, we see that the PS sprays differ from these trends. The 30 °C PS shows no trend regarding gap ratio. The 100 °C PS proximal also shows no trend, while 100 °C PS distal does show some relation to the gap ratio (Figure S8). However, both show a strong behavioral correlation to feature size, for the above-stated reason of smaller features being more difficult to target specifically. For the 100 °C PS, we can also see that the high-density, low-specificity data points are mainly derived from smaller features at distal locations. MG, on the other hand, has trends for both the gap ratio and the feature size but is relatively independent of distal/proximal positioning. As shown above, the dependence on the gap is strongly contributed to by a coincidence with the spray thickness gradient. However, we also see in Figure 4b that for the 20 μm/9× gap and 20 μm/3× gap, similar amounts of material are deposited, but there is a large discrepancy in density, demonstrating the metric bias that occurs for coatings trending toward indiscriminate behavior.

One way to think about this analysis is as a spectrum from metric-bias-dominated behavior to phenomenological-dominated behavior. Data from the simulated sample represent what we might expect from a conventional coating method such as spin coating, and it is completely dominated by the inherent biases in our metrics. The MG coating is relatively thin for most of the sample due to the thickness gradient, which is too strong to allow us to hypothesize any other effect for the gap. However, the MG is not completely indiscriminate. Repulsion from the masked region prevents larger features with larger gaps from blending, which is why we see a trend in Figure 7. If we were to increase the MG coating thickness via longer spray times, some of these larger features would likely spread more, and the high specificity values would decrease. However, we would also likely observe a transition to charged melt behavior before material on isolated features spreads far beyond the test pattern. PS sprays, on the other hand, are better able to accumulate charge, even if 100 °C PS can spread and dissipate the charge to some degree. The 100 °C PS distal is less targeted than 100 °C PS proximal and 30 °C PS and, therefore, shows metric bias to a slight degree (Figure S8). The 30 °C PS and 100 °C PS proximal act independent of metric bias, which provides evidence for SL behavior. Further, by considering the 30 °C PS features less than 240 μm, the specificity versus inverse density plot can be fit linearly (R = 0.88) with a slope of 1.49 ± 0.10 and a near-zero intercept of $3.03 \times 10^{-4} \pm 0.05$. As per Equation (4), the slope represents the average height of the features, which is measured directly as 1.51 ± 0.10 μm. This confirms that the SL effect for small features leads to uniform films. For the 240 μm features, the points are more scattered corresponding to the change in profile shape observed for larger features (Figure S6), and the slight increase in height seen in Figure 4a appears to have more of an impact here.

### 3.3. Feature Crosstalk

A separate figure of merit that is highly relevant to microfabrication is feature crosstalk—i.e., whether a deposition connects two adjacent features. This could be fatal or essential based on the application. For example, interdigitated electrodes have been used for flexible supercapacitors [27,28]. In this case, bridging the gap with a conductive material would short the device. However, if we were to use ESD for encapsulation, we would want to ensure that the entire grating is coated.

We create a new binary model profile with wider features to capture this scenario. The width of a single feature and gap is added, and the feature is expanded to fill 90% of this space (Figure 8a). Gratings that only include a single feature are excluded from this analysis. While specificity increases, only material bridging the space directly in the center of the gap is penalized. Examining the results (Figures 8b and S9) and cross-referencing with the profilometry scans (Figures S1–S3), we see that anything less than 99% has some features bridged by the coating. This gives us more concrete data for the limits of SLED. The 30 °C PS, below T$_g$, spray allows us to access features separated by gaps of 80–120 μm where 100 °C PS fails. We also see that the limit for 30 °C PS appears to be 60 μm gaps and below. Interestingly, this is irrelevant of feature size, telling us that feature size controls the specificity and the density to some degree, but ultimately, the amount of spreading is

relatively similar for all features. Therefore, the size of the gap is the only important factor in this regard.

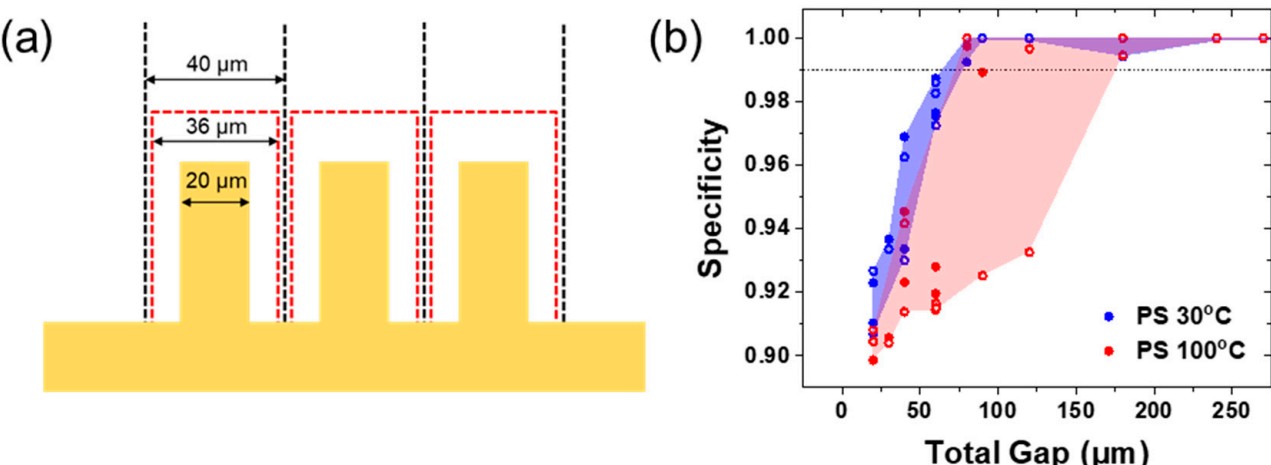

**Figure 8.** (**a**) Cross-section drawing of the expanded-feature-generated profile (red) relative to the actual feature (gold) and the total gap (black); (**b**) Specificity results for the expanded features.

While we would expect the general trends to hold for other sprays and substrate systems, we would expect the actual size limits to vary. As seen in Figure S11, the focusing behavior can vary based on the surrounding surface charge. Increasing the effect of focusing while still remaining in a SL effective flowrate regime could increase specificity. Another option would be to spray for shorter times, though this would decrease the density and risk non-uniformity due to thickness gradients in less developed films. In Figure S10, we show an example of another SL material, methylcellulose. While it appears to have targeted the features with higher fidelity, it is also undensified, so the difference might be due to post-spray flow. Within SL materials, behavior differences can likely occur due to dielectric constant, morphology, and solvent absorption, though we expect general trends compared to electrowetting and charged melt materials to remain the same.

## 4. Conclusions

In this study, we demonstrate that the mobility of electrosprayed materials strongly impacts the material's interaction with features of varying geometry. For a liquid—electrowetting material—represented by the MG, it is demonstrated that deposition is strongly, though not completely, controlled by the spray thickness gradient. This gradient is observed across the test pattern, even within a single grating, increasing both the measured heights and densities. However, the liquid is still charged and is, therefore, not completely indiscriminate. Accordingly, the effects of charge repulsion still allow for the high-specificity, high-density coatings seen on large features.

The 100 °C PS shows intermediate charged melt behavior between the SL regime and the completely liquid electrowetting regime. The higher temperature allows the material to spread, which alleviates charge buildup, increasing the density and decreasing the specificity, and this effect is more prominent on distal regions of the feature. However, there is still enough charge accumulation to prevent the formation of a strong thickness gradient and provide a more uniform coating, though the feature size appears to control the film thickness.

Finally, 30 °C PS is clearly representative of SL behavior. Its density and specificity do not trend with the model biases whatsoever, and there is relatively little difference between distal and proximal scans. Its specificity, especially for small features, is consistently much higher than any other material shown here, and there is lower variability. Its density is also relatively uniform and, on average, lower than that of 100 °C PS. Most interesting of all, it is found that while smaller features tend to have higher densities and lower

specificities, height is relatively constant, indicating that film growth is uniform across features. We also expect that other self-limiting materials would demonstrate this general behavior (Figure S10).

These results differ from our previous findings which showed that for small circular features with diameters ranging from 10 to 40 μm, wider features had thicker coatings, and all these coatings were much thicker than unpatterned electrosprays. This is more similar to what we observed here for 100 °C PS distal. We, therefore, postulate that geometry only impacts the height of non-SL materials. In line with this conclusion, it appears that for our previous results, the deposited material entered the charged-melt regime, likely from an excess accumulation of solvent, allowing for continued material growth that was further compounded by the effects of focusing. The evidence presented in this work suggests that we were able to mitigate this effect, likely through a combination of a lower flow rate, greater spray distance, and larger pattern area.

Finally, we show that the self-limiting material sprayed on these test patterns could target features separated by 80 μm gaps and above without bridging them. The fact that 80 μm is observed as the limit across feature sizes speaks to the material's ability to grow uniformly. However, we do expect this number to be system-dependent, and we predict that it is possible to target features separated by even smaller gaps in other systems. For example, it is likely possible that stronger focusing could be employed at the low flow rates used in this work without great detriment to the uniformity. However, in this case, we can use 80 μm to inform pattern design within this system, and we can use our analysis technique to rapidly assess new systems in the future. Furthermore, we can use the current data to guide the deployment of these materials under different applications. For example, to encapsulate with the melting gel material, we would likely use a moving stage or a focusing collar to ensure more uniform coverage of all features as has been conducted in the past [24]. On the other hand, the uniformity of the SL material on small features is a very promising result that will allow for controlled deposition of multilayered materials.

**Supplementary Materials:** The following supporting information can be downloaded at: https://www.mdpi.com/article/10.3390/coatings13030599/s1, Figure S1: All aligned profile data for the 30 °C PS; Figure S2: All aligned profile data for the 100 °C PS; Figure S3: All aligned profile data for the MG.; Figure S4: Visual representation of (top) $H_g$, (middle) $\rho_g$, and (bottom) $\sigma_g$; Figure S5: Profile scan of (a) the 120 μm feature of the MG test pattern and (b) the 20 μm features overlayed with the 240 μm features; Figure S6: Overlays of the 30, 60, and 240 μm feature profiles, standardized by width for (a) PS 30 °C Proximal and (b) PS 100 °C Proximal and Distal; Figure S7: Specificity versus density for all data sets; Figure S8: Specificity vs 1/Density with color maps denoting gap ratio for PS 100 °C distal-only; Figure S9: Expanded gap specificity by feature size. (a) 20 μm, (b) 30 μm, (c) 60 μm and (d) 120 μm; Figure S10: Microscope images of the 60 μm feature/$3\times$ gap for (a) blank test pattern, (b) MG, (c) PS 30 °C, and (d) PS 100 °C, and (e) methylcellulose; Figure S11: Pictures of the simulation cell for the electrostatic spray simulation.

**Author Contributions:** Conceptualization, J.P.S. and M.J.G.; methodology, J.P.S., M.J.G., M.A. and J.D.Z.; software, M.J.G.; investigation M.J.G.; formal analysis, M.J.G.; writing—original draft preparation, J.P.S. and M.J.G.; writing—review and editing, J.P.S., M.A. and J.D.Z.; resources, A.J., A.S. and M.A.; visualization, J.P.S. and M.J.G.; supervision, J.P.S., J.D.Z. and A.J.; project administration, J.P.S.; funding acquisition, J.P.S., J.D.Z. and A.J. All authors have read and agreed to the published version of the manuscript.

**Funding:** This work was partially funded by NSF Advanced Manufacturing Awards #1911518 and #1911509 and GeneOne Life Science.

**Institutional Review Board Statement:** Not applicable.

**Informed Consent Statement:** Not applicable.

**Data Availability Statement:** Full data are available on reasonable request to the corresponding author.

**Acknowledgments:** The authors would like to thank Zaynab Hazaveh for contributions to the processing of the profilometry data.

**Conflicts of Interest:** J.P.S. is an inventor on international patent application PCT/US20/33020 and its derivative regional patent applications that include aspects of self-limiting electrospray technology. The other authors declare no conflict of interest.

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
