# Peer review of "Microscale Templating of Materials across Electrospray Deposition Regimes"

_coatings, doi:10.3390/coatings13030599_

Round 1
Reviewer 1 Report
The paper is very interesting and deserve to be published. However, a SEM image each three spray regimes would be interesting to see. If the authors have, please add to the manuscript.
Author Response
We thank the reviewer for the very positive review. Unfortunately we do not have SEM images of these samples. However, we have several SEM images in our prior work, notably:
Lei et al. ACS Appl. Mater. Interfaces 2018, 10, 13, 11175–11188
We were able to add microscope images of the test patterns as Figure S10. This figure should be useful for providing further clarity regarding spray regime behavior.
Author Response
We thank the reviewer for their comments. The minor mistakes referenced by the reviewer were addressed through a thorough edit of the manuscript. The sections have been renumbered. Figure 5 has been properly explained. Equations in section 2.4 have been justified, and the rest of the manuscript has been reviewed for grammar and clarity.
Reviewer 3 Report
The ESD patterning on the micron-scale has been studied for decades, most typically through the use of insulating masks; there has been no comparative study of these phenomena across ESD spray regimes. Authors use test-patterns composed of gratings that range in both feature size and spacing to compare materials across regimes. The sprayed patterns were scanned using a profilometer, and the density, average height, and specificity were extracted. From these results, it is demonstrated that material deposited in the self-limiting regime shows the highest uniformity and specificity on small features as compared to electro wetting and charged melt sprays.
The publication covers basic research aimed at determining the factors determining the quality of the product obtained in the electrospray deposition process. The research covers one material - polystyrene and one solvent - 2-Butanone, which is important but very limited in terms of materials. The publication should include important information to what extent they obtained results are of a general nature, also for other materials. In electrospray deposition research, we have two philosophies, the first one consists in obtaining the material and then testing its properties, and the second one consists in looking for experimental conditions in order to obtain a material with predetermined physicochemical properties. Currently, electrospray deposition techniques are increasingly used to obtain a material with defined physicochemical properties. The presented work presents the study of the material whose properties were determined after the experiment. In the discussion, information should be added about what parameters are/may be crucial to obtain a material with the assumed properties before the experiment.
The publication, after adding a discussion on the possibility of generalizing the results to other materials and information on what parameters are/may be crucial for obtaining a material with the assumed properties before the experiment, will be of interest to a wide range of chemists and engineers.
Author Response
The publication, after adding a discussion on the possibility of generalizing the results to other materials and information on what parameters are/may be crucial for obtaining a material with the assumed properties before the experiment, will be of interest to a wide range of chemists and engineers.
We thank the reviewer for their excellent suggestion. We refer the reviewer to lines 60-62 regarding the crucial parameters. In general, amorphous polymeric materials below their Tg would be expected to show SL behavior. In response to the request for generalizability, we have included Figure S10 which shows the behavior of another SL material. We reference the figure in section 3.3:
In Figure S10 we show an example of another SL material, methylcellulose. While it appears to have targeted the features with higher fidelity, it is also undensified, so the difference might be due to post-spray flow. Within SL materials, behavior differences can likely occur due to dielectric constant, morphology, and solvent absorption, though we expect general trends compared to electrowetting and charged melt materials to remain the same.